# A mathematical model for cancer risk and accumulation of mutations caused by replication errors and external factors

**Kouki Uchinomiya** [ORCID] *, **Masanori Tomita**

Biology and Environmental Chemistry Division, Sustainable System Research Laboratory, Central Research Institute of Electric Power Industry, Komae, Tokyo, Japan

* u-kouki@criepi.denken.or.jp

## Abstract

Replication errors influence mutations, and thus, lifetime cancer risk can be explained by the number of stem-cell divisions. Additionally, mutagens also affect cancer risk, for instance, high-dose radiation exposure increases lifetime cancer risk. However, the influence of low-dose radiation exposure is still unclear because this influence, if any, is very slight. We can assess the minimal influence of the mutagen by virtually comparing the states with and without mutagen using a mathematical model. Here, we constructed a mathematical model to assess the influence of replication errors and mutagens on cancer risk. In our model, replication errors occur with a certain probability during cell division. Mutagens cause mutations at a constant rate. Cell division is arrested when the number of cells reaches the capacity of the cell pool. When the number of cells decreases because of cell death or other reasons, cells resume division. It was assumed that the mutations of cancer driver genes occur stochastically with each mutation and that cancer occurs when the number of cancer driver gene mutations exceeds a certain threshold. We approximated the number of mutations caused by errors and mutagens. Then, we examined whether cancer registry data on cancer risk can be explained only through replication errors. Although the risk of leukemia was not fitted to the model, the risks of esophageal, liver, thyroid, pancreatic, colon, breast, and prostate cancers were explained only by replication errors. Even if the risk was explained by replication errors, the estimated parameters did not always agree with previously reported values. For example, the estimated number of cancer driver genes in lung cancer was larger than the previously reported values. This discrepancy can be partly resolved by assuming the influence of mutagen. First, the influence of mutagens was analyzed using various parameters. The model predicted that the influence of mutagens will appear earlier, when the turnover rate of the tissue is higher and fewer mutations of cancer driver genes were necessary for carcinogenesis. Next, the parameters of lung cancer were re-estimated assuming the influence of mutagens. The estimated parameters were closer to the previously reported values. than when considering only replication errors. Although it may be useful to explain cancer risk by replication errors, it would be biologically more plausible to consider mutagens in cancers in which the effects of mutagens are apparent.

**Data Availability Statement:** Statistical data used in the manuscript can be accessed from the "graph database" in "latest cancer statistics" of Cancer

Information Service, National Cancer Center, Japan (National Cancer Registry, Ministry of Health, Labour and Welfare (https://ganjoho.jp/reg_stat/index.html). The direct URL to the "graph database" for each cancer and the values used there are shown below. "Graph database" of esophageal cancer: https://gdb.ganjoho.jp/graph_db/gdb1?dataType=30&graphId=106&totalTarget=40&_useLog=on&_stackedRaito=on&_showErrorBar=on&_useUnknownStage=on&smTypes=1&_smTypes=on&_smTypes=on&_smTypes=on&_smTypes=on&_smTypes=on&_smTypes=on&_smTypes=on&_smTypes=on&_smTypes=on&_smTypes=on&_smTypes=on&_smTypes=on&_smTypes=on&_smTypes=on&_smTypes=on&_smTypes=on&_smTypes=on&_smTypes=on&_smTypes=on&_smTypes=on&_smTypes=on&_smTypes=on&_smTypes=on&_smTypes=on&_smTypes=on&smType=4&year=2015&years=2015&_years=1&avgStep=&survivalAgeKbn=AAA&sexType=0&ageSybt=0&ageSt=009&ageEd=A85¤tAge=0&marumeAgeKbn=A85&stage=0&elapsedYears=5&_showBreastOnlyFemale=on&_showOnlyPrefectures=on&showGraph=Submit
The values of the data of esophageal cancer: https://gdb.ganjoho.jp/graph_db/gdb1?showData=&dataType=30&graphId=106&totalTarget=40&year=2015&years=2015&avgStep=&survivalAgeKbn=AAA&ageSybt=0&ageSt=009&ageEd=A85¤tAge=0&smTypes=1&smType=4&sexType=0&marumeAgeKbn=A85&stage=0&elapsedYears=5 "Graph database" of leukemia: https://gdb.ganjoho.jp/graph_db/gdb1?dataType=30&graphId=106&totalTarget=40&_useLog=on&_stackedRaito=on&_showErrorBar=on&_useUnknownStage=on&smTypes=1&_smTypes=on&_smTypes=on&_smTypes=on&_smTypes=on&_smTypes=on&_smTypes=on&_smTypes=on&_smTypes=on&_smTypes=on&_smTypes=on&_smTypes=on&_smTypes=on&_smTypes=on&_smTypes=on&_smTypes=on&_smTypes=on&_smTypes=on&_smTypes=on&_smTypes=on&_smTypes=on&_smTypes=on&_smTypes=on&_smTypes=on&_smTypes=on&_smTypes=on&smType=27&year=2015&years=2015&_years=1&avgStep=&survivalAgeKbn=AAA&sexType=0&ageSybt=0&ageSt=009&ageEd=A85¤tAge=0&marumeAgeKbn=A85&stage=0&elapsedYears=5&_showBreastOnlyFemale=on&_showOnlyPrefectures=on&showGraph=Submit
The values of the data of leukemia: https://gdb.ganjoho.jp/graph_db/gdb1?showData=&dataType=30&graphId=106&totalTarget=40&year=2015&years=2015&avgStep=&survivalAgeKbn=AAA&ageSybt=0&ageSt=009&ageEd=A85¤tAge=

## Introduction

Nordling showed that age-dependent cancer death-rate becomes a straight line on the doubly logarithmic plane [1]. This is mathematically formularized and called the multistage theory of carcinogenesis [2]. This theory assumes that some successive mutations cause carcinogenesis, and each mutation arises with a certain probability depending on time. Then, the logarithm of the incidence rate increases linearly with the logarithm of the time unit, such as age. The slope of the line on the doubly logarithmic plane is used for estimating the number of mutations that are necessary for carcinogenesis. It was estimated that six or seven successive mutations are necessary for the mortality of cancer if the probability of mutation remains constant throughout life. Nowadays, it is known that the number of driver mutations in the tumor has a certain distribution, and it can be smaller than seven [3]. In addition, a different mathematical model showed that the risk of chronic myeloid leukemia could be explained by a single mutation [4]. Although molecular genetic data and the mathematical model have been updated, the fundamental concept that one or more mutations cause carcinogenesis is predominant. One of the important points of the multistage theory is that mutations arise depending on time. In other words, the number of mutations increases in an age-dependent manner with the number of cell divisions. Tomasetti and Vogelstein showed that cancer risk variation is explained by the number of stem-cell divisions [5]. They estimated the total number of stem-cell divisions and investigated its correlation with the lifetime risk of different types of cancers. They found that the risk of cancers and the number of cell divisions were strongly correlated, implying that cancer driver gene mutations are caused by cell divisions. Therefore, based on the multistage theory of carcinogenesis, not only the lifetime risk of cancer but also age-dependent cancer risk might be explained by replication errors. A study focused on cell division and cancer risk to explain the risk of colorectal cancer assuming that the rate of mutation is constant [6]. If the division rate of cells is constant, the assumption would be reasonable. However, the rate of cell division can change with age [5, 7]. Especially, cell division during the development of tissues might affect cancer risk [5]. Therefore, it is prudent to formularize assuming that the rate of cell proliferation varies with age.

Even if cancer risk is explained by the number of cell divisions, some mutagens such as ionizing radiation and smoking influence it. A study showed that the number of stem-cell divisions is not correlated with radiation- or smoking-associated cancer risk [8]. For example, a life-span study of atomic bomb survivors showed that the risk of cancer mortality increased significantly for some tissues such as the stomach, lung, liver, colon, breast, and gallbladder [9]. Stem and early progenitor cells are recognized as important target cells in radiation carcinogenesis, and ionizing radiation contributes to the carcinogenic process by adding a few mutations [10]. Since the accident at the Fukushima Daiichi Nuclear Power Plant, the carcinogenic effects of low-dose and low-dose-rate radiation have become a major public concern in Japan. Although evaluation of the biological effects of low dose and/or low dose-rate ionizing radiation has important implications in radiation protection, the magnitude of cancer risk associated with low dose and/or low dose-rate radiation exposure is still unclear [11]. One of the difficulties in estimating the influence of a very low dose is that its impact on cancer risk is, if any, very low. To discuss a low influence of mutagens, the baseline of cancer risk must be considered. As mentioned above, the influence of replication errors on cancer risk may be the baseline. Considering the relationship between replication error and cancer risk using a simple model and comparing it with actual data will help identify the baseline. Examining how mutagens affect this baseline will thus help us evaluate it, even if the effect of a mutagen is very small.

Here, we constructed a simple mathematical model to assess the influence of cell division and mutagens on cancer risk. The model assumed cell division phases of growth and stability. In the growth phase, stem cells increase exponentially until the number of cells reaches the capacity of the stem-cell pool. The stable phase starts after that. Stem cells divide only when the cells are decreased in the stable phase. In these processes, mutations can arise due to replication errors and mutagens. Cancer risk was assumed to depend on mutations. Next, we examined whether age-dependent cancer risk can be explained only by replication errors through a comparison of the model with cancer registry data. Finally, the influences of mutagens are discussed qualitatively by varying the parameters, and lung cancer data were reanalyzed.

## Methods

### Overview of the mathematical model

Cells generally do not proliferate indefinitely. Some studies assume that stem cells frequently divide until the tissue is fully developed, and the frequency of cell division decreases after development [5, 12]. We assumed that tissues exhibit growth and stable phases. During the growth phase, stem cells divide at a constant rate until the total cell number is the complete size of the stem-cell pool. Then, the growth phase ends, and the stable phase starts. In the stable phase, stem cells do not divide as long as the number of cells can be maintained. Stem cells can be removed from the cell pool as a result of cell death and/or differentiation; however, we did not distinguish this here. When cells are removed, the remaining cells undergo division with a constant growth rate until the total cell number reaches the complete size of the cell pool. We assumed that mutations are classified into errors and lesions. Errors correspond to DNA replication errors, which can arise during stem-cell division. We assumed errors to occur according to a Poisson distribution with parameter $\lambda_1$ when a stem cell undergoes division. By contrast, lesions arise independently from cell division, thus reflecting effects of mutagens. We assumed that the number of lesions per cell per time depends on a Poisson distribution with parameter $\lambda_2$. When $\lambda_3$ is independent from time, there is a chronic influence of mutagens from the beginning. When discussing the temporary influence of mutagens, $\lambda_2$ should be the function of time. In this paper, we assumed that $\lambda_2$ is constant, and we examined the accumulation of errors and lesions. A schematic diagram is shown in Fig 1.

### Simulation using the Gillespie algorithm

As the number of stem cells is not constant, we used the Gillespie algorithm [13–15] to simulate changes in the numbers of mutations over time (S1 File). Let $n_{\{i,j\}}$ be the number of stem

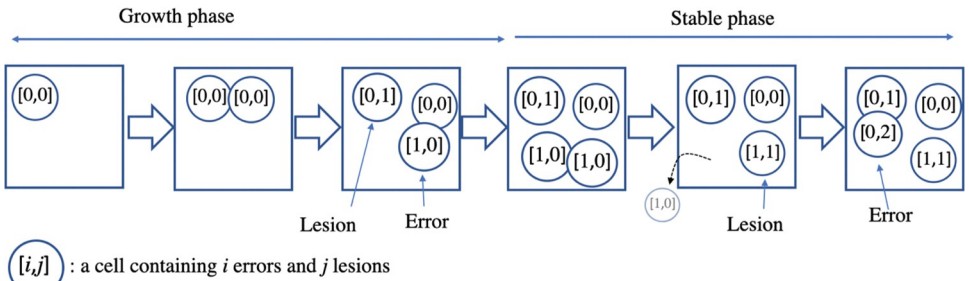

**Fig 1. Schematic diagram of the model.** This process is referred to as growth phase, followed by a stable phase. In the stable phase, cells can divide only when their number decreases, such as when they are eliminated. Errors can occur during division, and lesions occur with a certain probability at any time, regardless of division.

cells whose errors and lesions are $i$ and $j$, respectively. We express the condition of the stem-cell pool as a set of $n_{\{i,j\}}$ as follows: $(n_{\{0,0\}}, n_{\{1,0\}}, \ldots, n_{\{i,j\}}, \ldots, n_{\{I,J\}})$, where $I$ and $J$ are maximum numbers of errors and lesions, respectively. Although these limitations are necessary for the simulation, we can ignore their influence by setting sufficiently large $I$ and $J$. The condition of the tissue varies with stochastic events such as stem-cell division, formation of lesions, and cell removal. The chance of each event is proportional to its normalized form by the sum of the rates of all possible events [15]. First, we show the rate of each event in the growth phase. Stem cells divide at a rate $r$, and cell removal does not occur during the growth phase. As the maximum number of errors is $I$, the rate at which a stem cell carries $i$ errors and $j$ lesions and divides with $k$ errors is

$$\left(n_{\{0,0\}}, n_{\{1,0\}}, \ldots, n_{\{i,j\}}, \ldots, n_{\{I,J\}}\right) \rightarrow \left(n_{\{0,0\}}, n_{\{1,0\}}, \ldots, n_{\{i+k,j\}} + 1, \ldots, n_{\{I,J\}}\right) = \frac{\lambda_1^k e^{-\lambda_1}}{k!} r n_{\{i,j\}} \text{ if } k \leq I - i, \quad (1)$$

$$\left(n_{\{0,0\}}, n_{\{1,0\}}, \ldots, n_{\{i,j\}}, \ldots, n_{\{I,J\}}\right) \rightarrow \left(n_{\{0,0\}}, n_{\{1,0\}}, \ldots, n_{\{I,j\}} + 1, \ldots, n_{\{I,J\}}\right) = \sum_{k > I-i}^{\infty} \frac{\lambda_1^k e^{-\lambda_1}}{k!} r n_{\{i,j\}} \text{ if } k > I - i. (2)$$

$X \rightarrow Y$ represents the rate of the changing the condition from $X$ to $Y$. Cells can also suffer lesions without cell division. The rate at which a cell having $i$ errors and $j$ lesions and suffers $l$ lesions is written as follows:

$$\left(n_{\{0,0\}}, n_{\{1,0\}}, \ldots, n_{\{i,j\}}, \ldots, n_{\{i,j+l\}}, \ldots, n_{\{I,J\}}\right)$$
$$\rightarrow \left(n_{\{0,0\}}, n_{\{1,0\}}, \ldots, n_{\{i,j\}} - 1, \ldots, n_{\{i,j+l\}} + 1, \ldots, n_{\{I,J\}}\right) = \frac{\lambda_2^l e^{-\lambda_2}}{l!} n_{\{i,j\}} \text{ if } l \leq J - j, \quad (3)$$

$$\left(n_{\{0,0\}}, n_{\{1,0\}}, \ldots, n_{\{i,j\}}, \ldots, n_{\{i,J\}}, \ldots, n_{\{I,J\}}\right)$$
$$\rightarrow \left(n_{\{0,0\}}, n_{\{1,0\}}, \ldots, n_{\{i,j\}} - 1, \ldots, n_{\{i,J\}} + 1, \ldots, n_{\{I,J\}}\right) = \sum_{l > J-j}^{\infty} \frac{\lambda_2^l e^{-\lambda_2}}{l!} n_{\{i,j\}} \text{ if } l > J - j. \quad (4)$$

Then, the sum of the rates of all possible events in the growth phase, $\Gamma_g$, is

$$\Gamma_g = \sum_{i=0}^{I} \sum_{j=0}^{J} \left[ \sum_{k=0}^{I-i} \frac{\lambda_1^k e^{-\lambda_2}}{k!} r n_{\{i,j\}} + \sum_{k > I-i}^{\infty} \frac{\lambda_1^k e^{-\lambda_2}}{k!} r n_{\{i,j\}} + \sum_{l=0}^{J-j} \frac{\lambda_2^l e^{-\lambda_2}}{l!} n_{\{i,j\}} + \sum_{l > J-j}^{\infty} \frac{\lambda_2^l e^{-\lambda_2}}{l!} n_{\{i,j\}} \right]$$
$$= \sum_{i=0}^{I} \sum_{j=0}^{J} [r n_{\{i,j\}} + n_{\{i,j\}}]. \quad (5)$$

Denoting the probability that condition $X$ changes to $Y$ as $\Pr[X \rightarrow Y]$, we can write the probability of each event as follows:

$$\Pr[(n_{\{0,0\}}, n_{\{1,0\}}, \ldots, n_{\{i,j\}}, \ldots, n_{\{I,J\}}) \rightarrow (n_{\{0,0\}}, n_{\{1,0\}}, \ldots, n_{\{i+k,j\}} + 1, \ldots, n_{\{I,J\}})] = \frac{1}{\Gamma_g} \frac{\lambda_1^k e^{-\lambda_1}}{k!} r n_{\{i,j\}} \text{ if } k \leq I - i, \quad (6)$$

$$\Pr[(n_{\{0,0\}}, n_{\{1,0\}}, \ldots, n_{\{i,j\}}, \ldots, n_{\{I,J\}}) \rightarrow (n_{\{0,0\}}, n_{\{1,0\}}, \ldots, n_{\{I,j\}} + 1, \ldots, n_{\{I,J\}})] = \frac{1}{\Gamma_g} \sum_{k > I-i}^{\infty} \frac{\lambda_1^k e^{-\lambda_1}}{k!} r n_{\{i,j\}} \text{ if } k > I - i, (7)$$

$$\Pr[(n_{\{0,0\}}, n_{\{1,0\}}, \ldots, n_{\{i,j\}}, \ldots, n_{\{i,j+l\}}, \ldots, n_{\{I,J\}})$$
$$\rightarrow (n_{\{0,0\}}, n_{\{1,0\}}, \ldots, n_{\{i,j\}} - 1, \ldots, n_{\{i,j+l\}} + 1, \ldots, n_{\{I,J\}})] = \frac{1}{\Gamma_g} \frac{\lambda_2^l e^{-\lambda_2}}{l!} n_{\{i,j\}} \text{ if } l \leq J - j, \quad (8)$$

$$\Pr[(n_{\{0,0\}}, n_{\{1,0\}}, \dots, n_{\{i,j\}}, \dots, n_{\{I,J\}}) \rightarrow (n_{\{0,0\}}, n_{\{1,0\}}, \dots, n_{\{i,j\}} - 1, \dots, n_{\{i,J\}} + 1, \dots, n_{\{I,J\}})]$$

$$= \frac{1}{\Gamma_g} \sum_{l > J-j}^{\infty} \frac{\lambda_2^l e^{-\lambda_2}}{l!} n_{\{i,j\}} \quad \text{if } l > J - j. \tag{9}$$

Let $N_L$ be the capacity of the stem-cell pool. In the stable phase, cells do not divide when there are $N_L$ stem cells, and cells may be removed. We denote $a_{\{i,j\}}$ as the rate of cell division of a stem cell, which carries $i$ errors and $j$ lesions in the stable phase. As the division rate is 0 when there are $N_L$ cells,

$$a_{\{i,j\}} = \begin{cases} 0 & \text{if } N = N_L \\ a_{d\{i,j\}} & \text{if } N < N_L \end{cases}, \tag{10}$$

where $N$ is the total cell number, which is defined as

$$N = \sum_{i=0}^{I} \sum_{j=0}^{J} n_{\{i,j\}}. \tag{11}$$

When the total cell number is smaller than $N_L$, the stem cells divide at a rate of $a_{d\{i,j\}}$. Similar to the growth phase, the rate at which a stem cell, which carries $i$ errors and $j$ lesions, divides with $k$ errors is

$$\left(n_{\{0,0\}}, n_{\{1,0\}}, \dots, n_{\{i,j\}}, \dots, n_{\{I,J\}}\right) \rightarrow \left(n_{\{0,0\}}, n_{\{1,0\}}, \dots, n_{\{i+k,j\}} + 1, \dots, n_{\{I,J\}}\right) = \frac{\lambda_1^k e^{-\lambda_1}}{k!} a_{\{i,j\}} n_{\{i,j\}} \quad \text{if } k \leq I - i, \tag{12}$$

$$\left(n_{\{0,0\}}, n_{\{1,0\}}, \dots, n_{\{i,j\}}, \dots, n_{\{I,J\}}\right) \rightarrow \left(n_{\{0,0\}}, n_{\{1,0\}}, \dots, n_{\{i,j\}} + 1, \dots, n_{\{I,J\}}\right) = \sum_{k>I-i}^{\infty} \frac{\lambda_1^k e^{-\lambda_1}}{k!} a_{\{i,j\}} n_{\{i,j\}} \quad \text{if } k > I - i. \tag{13}$$

The rate of cell removal is

$$(n_{\{0,0\}}, n_{\{i,j\}}, \dots, n_{\{I,J\}}) \rightarrow (n_{\{0,0\}}, n_{\{i,j\}} - 1, \dots, n_{\{I,J\}}) = b_{\{i,j\}} n_{\{i,j\}}, \tag{14}$$

where $b_{\{i,j\}}$ is the removal rate of a stem cell carrying $i$ errors and $j$ lesions.

As the mutations caused by lesions is the same as that during the growth phase the sum of the rates of all possible events in the stable phase, $\Gamma_s$, is

$$\Gamma_s = \sum_{i=0}^{I} \sum_{j=0}^{J} \left[ \sum_{k=0}^{I-i} \frac{\lambda_1^k e^{-\lambda_1}}{k!} a_{\{i,j\}} n_{\{i,j\}} + \sum_{k>I-i}^{\infty} \frac{\lambda_1^k e^{-\lambda_1}}{k!} a_{\{i,j\}} n_{\{i,j\}} + b_{\{i,j\}} n_{\{i,j\}} + \sum_{l=0}^{J-j} \frac{\lambda_2^l e^{-\lambda_2}}{l!} n_{\{i,j\}} + \sum_{l>J-j}^{\infty} \frac{\lambda_2^l e^{-\lambda_2}}{l!} n_{\{i,j\}} \right]$$

$$= \sum_{i=0}^{I} \sum_{j=0}^{J} [a_{\{i,j\}} n_{\{i,j\}} + b_{\{i,j\}} n_{\{i,j\}} + n_{\{i,j\}}]. \tag{15}$$

Then, the probability of each event in the stable phase can be denoted as follows:

$$\Pr[(n_{\{0,0\}}, n_{\{1,0\}}, \dots, n_{\{i,j\}}, \dots, n_{\{I,J\}}) \rightarrow (n_{\{0,0\}}, n_{\{1,0\}}, \dots, n_{\{i+k,j\}} + 1, \dots, n_{\{I,J\}})]$$

$$= \frac{1}{\Gamma_s} \frac{\lambda_1^k e^{-\lambda_1}}{k!} a_{\{i,j\}} n_{\{i,j\}} \quad \text{if } k \leq I - i, \tag{16}$$

$$\Pr[(n_{\{0,0\}}, n_{\{1,0\}}, \dots, n_{\{i,j\}}, \dots, n_{\{I,J\}}) \rightarrow (n_{\{0,0\}}, n_{\{1,0\}}, \dots, n_{\{I,j\}} + 1, \dots, n_{\{I,J\}})]$$

$$= \frac{1}{\Gamma_s} \sum_{k>I-i}^{\infty} \frac{\lambda_1^k e^{-\lambda_1}}{k!} a_{\{i,j\}} n_{\{i,j\}} \quad \text{if } k > I - i, \tag{17}$$

on&_smTypes=on&_smTypes=
on&_smTypes=on&_smTypes=on&_smTypes=
on&_smTypes=on&_smTypes=on&_smTypes=
on&_smTypes=on&_smTypes=on&_smTypes=
on&_smTypes=on&_smTypes=on&_smTypes=
on&_smTypes=on&_smTypes=on&_smTypes=
on&smType=20&year=
2015&years=2015&_years=1&avgStep=
&survivalAgeKbn=AAA&sexType=0&ageSybt=
0&ageSt=009&ageEd=A85¤tAge=
0&marumeAgeKbn=A85&stage=0&elapsedYears=
5&_showBreastOnlyFemale=on&_
showOnlyPrefectures=on&showGraph=Submit
The values of the data of prostate cancer: https://
gdb.ganjoho.jp/graph_db/gdb1?showData=
&dataType=30&graphId=106&totalTarget=
40&year=2015&years=2015&avgStep=
&survivalAgeKbn=AAA&ageSybt=0&ageSt=
009&ageEd=A85¤tAge=0&smTypes=1&smType=
20&sexType=0&marumeAgeKbn=A85&stage=
0&elapsedYears=5 The authors confirm that others
would be able to access or request these data in
the same manner as the authors. The authors did
not have any special access or request privileges
that others would not have.

**Funding:** KU was supported by JSPS KAKENHI
Grant Number JP 20K19972. The funders had no
role in study design, data collection and analysis,
decision to publish, or preparation of the
manuscript.

**Competing interests:** The authors have declared
that no competing interests exist.

$$\Pr[(n_{\{0,0\}}, n_{\{1,0\}}, \ldots, n_{\{i,j\}}, \ldots, n_{\{i,j+l\}}, \ldots, n_{\{I,J\}})$$

$$\rightarrow (n_{\{0,0\}}, n_{\{1,0\}}, \ldots, n_{\{i,j\}} - 1, \ldots, n_{\{i,j+l\}} + 1, \ldots, n_{\{I,J\}})] = \frac{1}{\Gamma_s} \frac{\lambda_2{}^l e^{-\lambda_2}}{l!} n_{\{i,j\}} \text{ if } l \leq J - j, \quad (18)$$

$$\Pr[(n_{\{0,0\}}, n_{\{1,0\}}, \ldots, n_{\{i,j\}}, \ldots, n_{\{i,J\}}, \ldots, n_{\{I,J\}}) \rightarrow (n_{\{0,0\}}, n_{\{1,0\}}, \ldots, n_{\{i,j\}} - 1, \ldots, n_{\{i,J\}} + 1, \ldots, n_{\{I,J\}})]$$

$$= \frac{1}{\Gamma_s} \sum_{l > J - j}^{\infty} \frac{\lambda_2{}^l e^{-\lambda_2}}{l!} n_{\{i,j\}} \text{ if } l > J - j, \quad (19)$$

$$\text{and } \Pr[(n_{\{0,0\}}, n_{\{i,j\}}, \ldots, n_{\{I,J\}}) \rightarrow (n_{\{0,0\}}, n_{\{i,j\}} - 1, \ldots, n_{\{I,J\}})] = \frac{1}{\Gamma_s} b_{\{i,j\}} n_{\{i,j\}}. \quad (20)$$

Although $a_{\{i,j\}}$ and $b_{\{i,j\}}$ can depend on the number of mutations if mutations affect the character of cells, we assumed that those parameters are independent of the number of mutations, $a = a_{\{i,j\}}$, $b = b_{\{i,j\}}$, in the following analysis, for simplicity.

## Multi-stage carcinogenesis model

Based on the multistage carcinogenesis theory, we assumed that cancer arises through successive mutations. We did not distinguish between mutations arising from errors or lesions and assumed that mutations of $g$ cancer driver genes cause carcinogenesis. Strictly, accumulation of mutations should be considered for each stem cell, however, we considered the average number of mutations per cell, for simplicity. The total numbers of errors and lesions at time $t$ in the tissue were calculated, respectively, as follows:

$$E(t) = \sum_{i=0}^{\infty} i n_{\{i,j\}}(t), \quad (21)$$

$$W(t) = \sum_{j=0}^{\infty} j n_{\{i,j\}}(t), \quad (22)$$

where the number of stem cells that have $i$ errors and $j$ lesions at time $t$ is denoted as $n_{\{i,j\}}(t)$. The average number of mutations in a stem cell is

$$m(t) = \frac{E(t) + W(t)}{N(t)}, \quad (23)$$

where $N(t)$ is the total number of stem cells at time $t$. The probability of having $g$ or more mutations of cancer driver genes equals one minus the probability of the number of mutations smaller than $g$. Let $p$ be the probability that one cancer driver gene mutation occurs per mutation. By using the cumulative distribution function of the binomial distribution, when there are $m(t)$ mutations, the cumulative cancer risk is

$$G(t) = 1 - \sum_{k=0}^{g-1} \binom{m(t)}{k} p^k (1-p)^{m(t)-k}. \quad (24)$$

The second term is the probability that the number of cancer driver gene mutations is smaller than $g$. If $m(t)$ is sufficiently high and $p$ is sufficiently low, binomial distribution can be

approximated using the Poisson distribution. Then, $G$ can be described as follows:

$$G(t) = 1 - \sum_{k=0}^{g-1} \frac{(pm(t))^k e^{-pm(t)}}{k!}. \tag{25}$$

In the following, we use Eq (25) for simplifying the calculation.

## Results

### Simulation and approximation of the accumulation of errors and lesions

First, the accumulation of errors and lesions was simulated using the Gillespie algorithm. An example of the simulation is shown in Fig 2. The number of stem cells increases exponentially in the growth phase until it reaches $N_L$ (Fig 2A). When the number of stem cells reaches $N_L$,

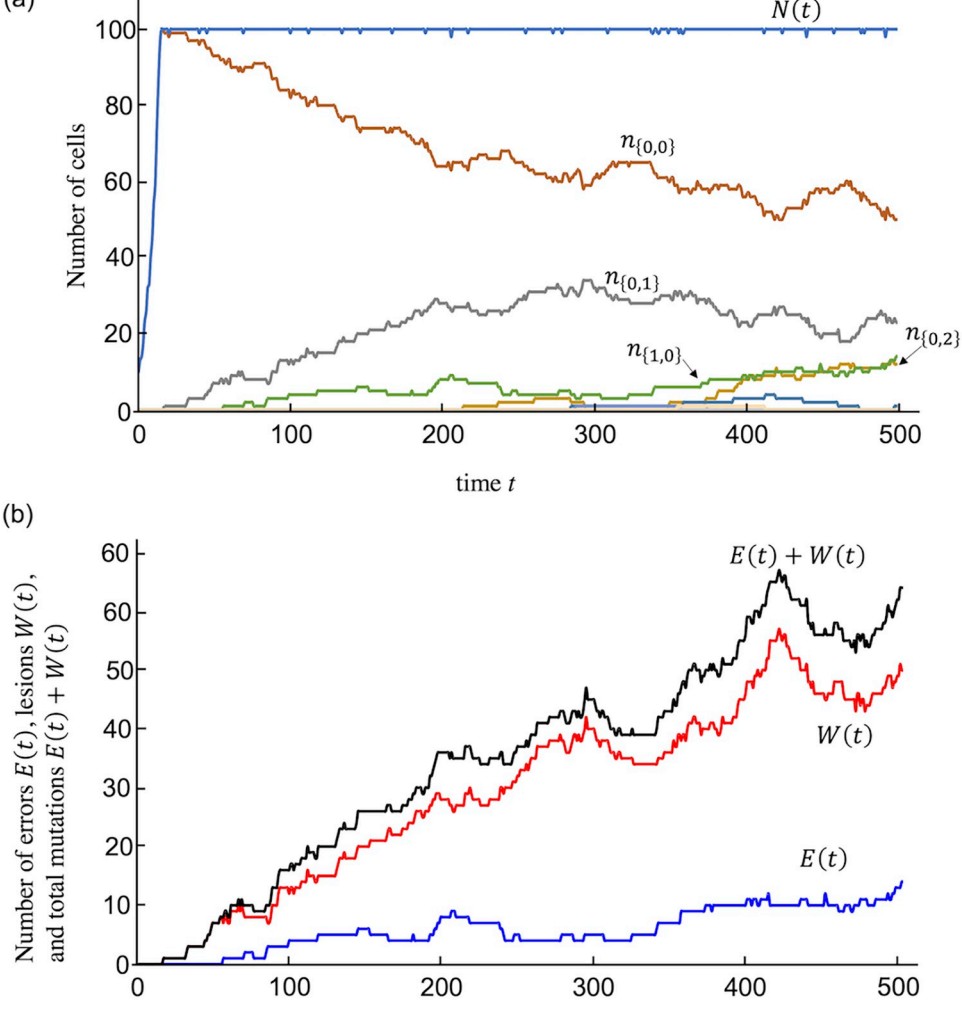

**Fig 2. Results of a simulation by the Gillespie algorithm.** (a) Time course of the number of cells. The label $n_{\{i,j\}}$ is the number of cells carrying $i$ errors and $j$ lesions. Total number of cells are denoted as $N(t)$. (b) Time course of the total number of errors $E(t)$ and lesions $W(t)$. Parameter values are $N_0 = 10$, $N_L = 100$, $r = 0.15$, $a_d = 0.16$, $b = 0.01$, $\lambda_1 = 0.01$, and $\lambda_2 = 0.001$. The results were output when reaching a certain number of divisions and were combined.

the growth phase ends, and the stable phase starts. When stem cells are removed, cell division resumes (Fig 2A). The time series of the number of errors and lesions are shown in Fig 2B. Then, we approximate the time course of the accumulation of errors and lesions to discuss cancer risk.

We consider the average accumulated mutations in the tissue. In the growth phase, stem cells increase with the growth rate $r$. Setting the initial stem-cell number to $N_0$, the number of cell divisions at time $t$ is $d$, and the following equation holds:

$$N_0\exp[rt] = N_0 + d. \tag{26}$$

Rearranging this equation, we can express $d$ as the function of time as follows:

$$d = N_0(\exp[rt] - 1). \tag{27}$$

As the average number of errors per cell division is $\lambda_1$, the number of errors, $E$, in the growth phase is

$$E = \lambda_1 d = \lambda_1 N_0(\exp[rt] - 1). \tag{28}$$

By contrast, the average number of lesions per stem cell per time is $\lambda_2$. The number of stem cells at time $t$ is $N_0 \exp[rt]$ in the growth phase. These mean that the increase rate of $W$ is $\lambda_2 N_0 \exp[rt]$. Considering there is no lesion in the initial state,

$$\frac{dW}{dt} = \lambda_2 N_0\exp[rt] \rightarrow W = \frac{\lambda_2 N_0}{r}(\exp[rt] - 1). \tag{29}$$

From Eq (27), $t$ can be written as the function of $d$ as follows:

$$t = \frac{1}{r}\log\left(\frac{d}{N_0} + 1\right). \tag{30}$$

Substituting Eq (30) into Eq (29) the number of lesions can be written as the function of $d$ as follows:

$$W = \frac{\lambda_2 N_0}{r}(\exp[rt] - 1) = \frac{\lambda_2 N_0}{r}\left(\left(\frac{d}{N_0} + 1\right) - 1\right) = \frac{\lambda_2}{r}d. \tag{31}$$

In the stable phase, the total number of cell divisions, $d$, is the summation of the total number of cell divisions in the growth and stable phases. The former is trivially calculated as $N_L - N_0$. To estimate the number of cell divisions in the stable phase, the waiting time until cell division needs to be taken into account. In the stable phase, stem cells do not divide when the number of stem cells is $N_L$. The waiting time until cell division is the summation of waiting time until cell removal and cell division. As the rate of cell removal is $b$, the waiting time until cell removal is $1/bN_L$. In this analysis, we assume the blank of a stem cell in the stem-cell pool is filled before another cell is removed. Then, the waiting time until cell division is approximated as $1/a(N_L-1)$. Therefore, the waiting time until cell division in the stable phase is calculated as follows:

$$\frac{1}{bN_L} + \frac{1}{a(N_L - 1)} = \frac{a(N_L - 1) + bN_L}{abN_L(N_L - 1)}. \tag{32}$$

Letting $d_T$ be the number of cell divisions during $T$ in the stable phase, the following equation holds:

$$d_T = \frac{T}{\frac{a(N_L - 1) + bN_L}{abN_L(N_L - 1)}}. \tag{33}$$

When growth phase ends at $t'$, the total number of cell divisions at $t > t'$ is

$$N_L - N_0 + d_{t-t'} = N_L - N_0 + \frac{t - t'}{\frac{a(N_L - 1) + bN_L}{abN_L(N_L - 1)}} == N_L - N_0 + \frac{(t - t')abN_L(N_L - 1)}{a(N_L - 1) + bN_L}. \tag{34}$$

As the growth phase ends when the total stem-cell number reaches $N_L$, $t'$ can be calculated as follows:

$$N_L = N_0 \exp[rt'] \rightarrow t' = \frac{1}{r}\log\frac{N_L}{N_0}. \tag{35}$$

Substituting this equation into Eq (34), the total number of cell divisions $d$ at $t > t'$ is

$$d = N_L - N_0 + \frac{\left(t - \frac{1}{r}\log\frac{N_L}{N_0}\right)abN_L(N_L - 1)}{a(N_L - 1) + bN_L}. \tag{36}$$

Then, the number of errors, $E$, in the stable phase is

$$E = d\lambda_1 = \lambda_1\left\{N_L - N_0 + \frac{\left(t - \frac{1}{r}\log\frac{N_L}{N_0}\right)abN_L(N_L - 1)}{a(N_L - 1) + bN_L}\right\}. \tag{37}$$

Considering the number of lesions in the growth phase, it is convenient to express $t$ as a function of $d$. From Eq (36), the relationship between $d$ and $t$ in the stable stage is

$$t = \frac{1}{r}\log\frac{N_L}{N_0} + \frac{(d - N_L + N_0)(a(N_L - 1) + bN_L)}{abN_L(N_L - 1)}. \tag{38}$$

Let $W'$ be the number of lesions at the end of the growth phase i.e. $W$ at $t = t'$. From Eqs (31) and (35), $W'$ can be calculated as

$$W' = \frac{\lambda_2 N_0}{r}\left(\exp\left[r\left(\frac{1}{r}\log\frac{N_L}{N_0}\right)\right] - 1\right) = \frac{\lambda_2}{r}(N_L - N_0). \tag{39}$$

The number of lesions emerged in the stable phase is expressed as the product of $N_L$, $\lambda_2$, and the time elapsed since entering the stable phase. The number of lesions in the stable phase is

$$W = N_L\lambda_2(t - t') + W' = N_L\lambda_2\left\{\frac{(d - N_L + N_0)(a(N_L - 1) + bN_L)}{abN_L(N_L - 1)}\right\} + \frac{\lambda_2}{r}(N_L - N_0). \tag{40}$$

Fig 3 shows a comparison between the results of simulations and approximations. The approximations explain the simulation well. Next, we discuss cancer risk using these approximations.

## Fitting to cancer registry data

We consider the cumulative risk of cancer as mutations accumulate during the lifetime. If cancer risk is explained by stem-cell divisions, the risk should be explained only by replication

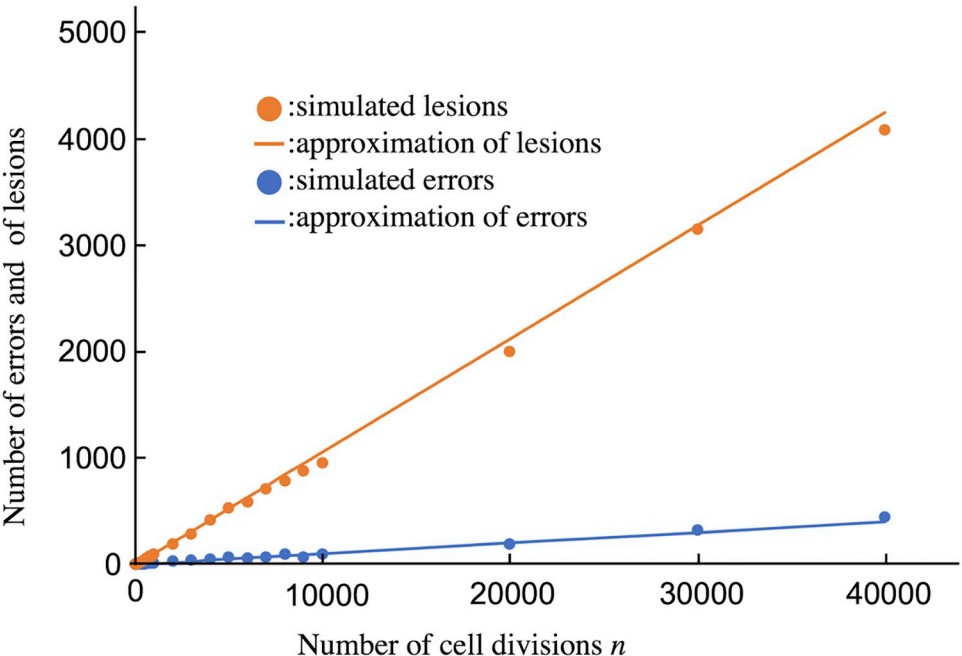

**Fig 3. Comparison of simulation and analytical approximations.** The horizontal line indicates the number of cell divisions, and the vertical line indicates the number of errors and lesions. Dots are the average values of 50-fold simulations. Lines are the approximations in Eqs 28, 29, 37, and 40. Blue and orange lines show errors and lesions, respectively. Parameter values are $N_0 = 10$, $N_L = 100$, $r = 0.15$, $a_d = 0.16$, $b = 0.01$, $\lambda_1 = 0.01$, and $\lambda_2 = 0.001$.

errors. In other words, the presented model can fit cancer registry data of cancer risk even if $\lambda_2 = 0$. We applied our model to cancer registry data of cumulative cancer incidence risk in 2015 obtained from the Cancer Information Service, National Cancer Center, Japan (Ministry of Health, Labour and Welfare, National Cancer Registry), accessible under https://ganjoho.jp/reg_stat/index.html. We assumed that a tissue originates from only a single stem cell and that the number of stem cells reaches $N_L$ at the age of 18, i.e., the growth phase continues until the age of 18. Therefore, the rate of cell division in the growth phase $r$ is

$$N_L = N_0 \exp[18r] \Longrightarrow r = \frac{1}{18} \log \frac{N_L}{N_0} = \frac{1}{18} \log N_L. \qquad (41)$$

The dimension of $r$ is cell divisions per year in this formula. When we assume that the cell division occurs immediately after a stem cell is removed, $a \gg b$, the waiting time of cell division in the stable phase, as in Eq (32), is $1/bN_L$. This means that the proliferation rate in the stable phase equals the cell-removal rate $b$. The number of stem cells in the tissue and that caused by the divisions of each stem cell per year are estimated in previous studies [5, 7] (Table 1). As we assume $\lambda_2 = 0$, we should estimate $\lambda_1$, $p$, and $g$. Explicitly denoting $d$ as a function of $t$ as $d(t)$, from Eqs (28) and (37), $pm(t) = p\lambda_1 d(t)$ if $\lambda_2 = 0$. Therefore, $\lambda_1 p$ is regarded as a single parameter in Eq (25) when $\lambda_2 = 0$. This parameter can be understood as the average number of cancer driver genes mutations per cell division. Therefore, we have to estimate the values of $\lambda_1 p$ and $g$. Although $\lambda_1 p$ is a real number, $g$ is an integer. We varied the value of $g$ and calculated the least-square value with $\lambda_1 p$. Then, we chose a parameter set of $\lambda_1 p$ and $g$ that made the least-square value minimum (S1 Fig). The curves of $G(t)$ were fitted to cancer registry data of esophageal cancer (Fig 4A), leukemia (Fig 4B), liver cancer (Fig 4C), lung cancer (Fig 4D), thyroid cancer (Fig 4E), pancreatic cancer (Fig 4F), colon cancer (Fig 4G), breast cancer (Fig 4H), and

**Table 1. List of parameters for the fitting to cancer registry data shown in Fig 4.**

| | Number of divisions of each stem cell per year ($b$) | Number of normal stem cells in tissue origin ($N_L$) | Cell division rate in growth phase ($r = \frac{\log N_L}{18}$) |
|---|---|---|---|
| Esophageal cancer | 33.2* | $6.6528 \times 10^6$* | 0.87 |
| Leukemia | 12* | $1.35 \times 10^8$* | 1.04 |
| Liver cancer | 0.91* | $3.01 \times 10^9$* | 1.21 |
| Lung cancer | 0.07* | $1.22 \times 10^9$* | 1.16 |
| Thyroid cancer | 0.087* | $6.5 \times 10^7$* | 1.00 |
| Pancreatic cancer | 1* | $4.18 \times 10^9$* | 1.23 |
| Colon cancer | 73* | $2.00 \times 10^8$* | 1.06 |
| Breast cancer | 4.32** | $8.7 \times 10^9$** | 1.27 |
| Prostate cancer | 2.99** | $2.1 \times 10^8$** | 1.06 |

*Values from [5]

**Values from [7]

prostate cancer (Fig 4I). The curve was not fitted to the risk of leukemia because the risk increases from a young age (Fig 4B). Except for leukemia, the curve was fitted to the data. This implies that age-dependent cancer risk can be explained well by replication errors. However, it should be noted that the estimated parameter values do not always agree with the values expected from the previously reported values, as elaborated in the Discussion. For example, $g$ and $\lambda_1 p$ of male lung cancer are very large. It is well known that smoking increases lung cancer risk. Therefore, it may be possible to reduce this discrepancy by considering the influence of mutagens.

## Influence of lesions on carcinogenesis

Finally, we provide a qualitative hypothesis for the influence of lesions on cancer risk by analyzing the model using different parameters. We used risk difference (RD) as a quantitative measure for the influence of lesions [16]. Let $G_1(t)$ and $G_0(t)$ be the cancer risks at time $t$ in the case of $\lambda_2 \neq 0$ and $\lambda_2 = 0$, respectively. Then, RD at time $t$ is defined as

$$RD(t) = G_1(t) - G_0(t). \tag{42}$$

The other measure called, risk ratio (RR), which is defined as $RR = G_1/G_0$, is widely used in the field of public health [16]. We, however, consider only RD because $G_0$ can be 0 in this theoretical analysis. Since the absolute value of RD depends on unknown biological parameters, such as sensitivity to mutagens, we assumed some parameter sets and obtained qualitative results. The typical shape of *RD* is shown in Fig 5. *RD* is very low at the beginning and after a sufficiently long time. Even if there are mutagens, the number of cancer driver gene mutations is not sufficient for carcinogenesis in the beginning. As a result, *RD* is low. By contrast, after a sufficiently long time, most cancers are caused by replication errors, and hence, *RD* is low. The largest values of *RD* are observed when $t$ is moderate. When $\lambda_2$ is large, the start of the increase is early, and the peak of *RD* is large (Fig 5A). This means that the greater the influence of mutagens, the earlier and greater the influence on *RD*. This result is intuitive. The biological features of the tissues might affect the influence of the mutagens. When the turnover of the stem cells is less frequent in the stable phase, i.e., $b$ is low, the increase of *RD* is late, and the maximum value is large (Fig 5B). In this case, the accumulation of errors is slow as the cell division is less frequent. Then, cancer driver gene mutations are relatively unlikely to occur even if there are

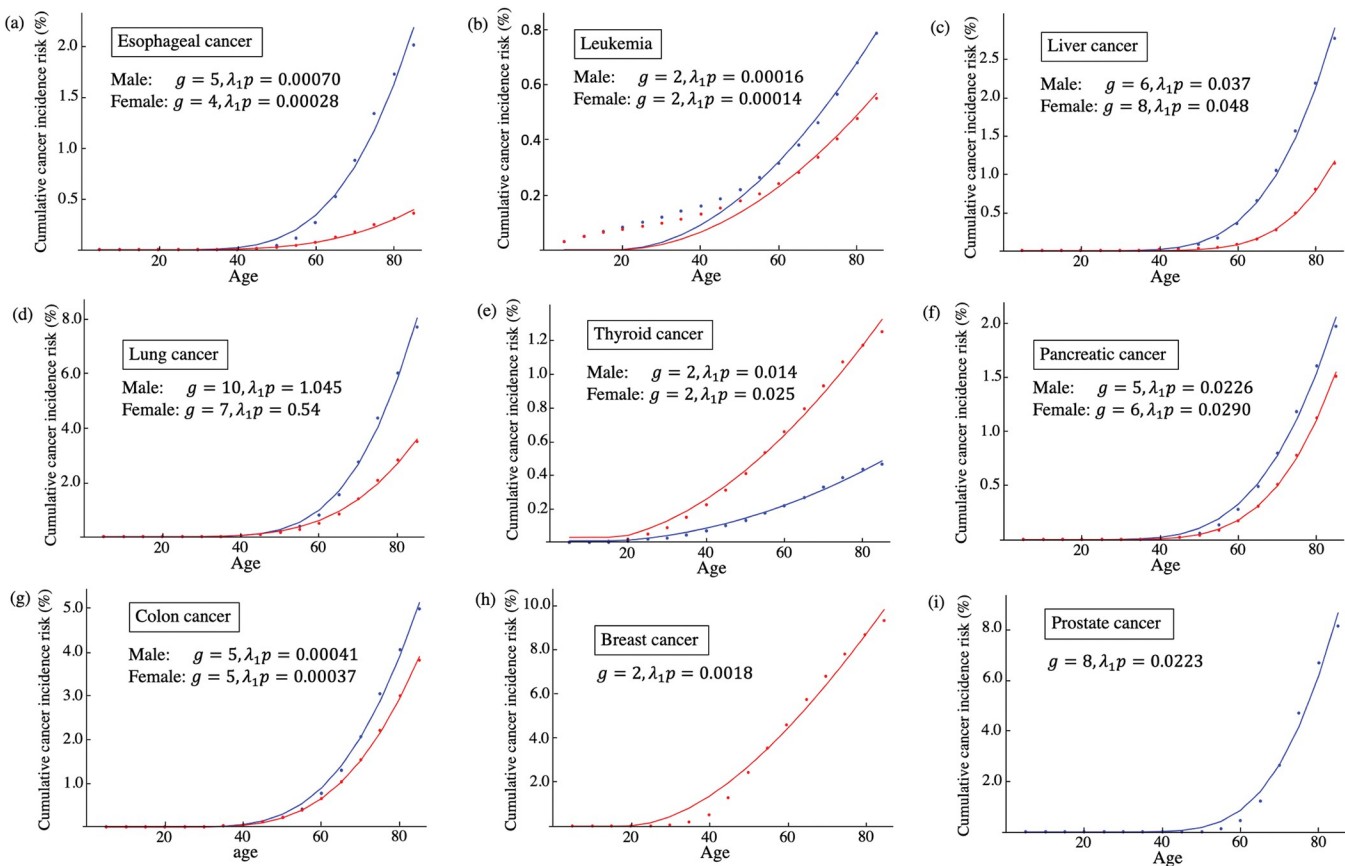

**Fig 4. Comparison of model and cancer registry data of age-dependent cancer risk.** Cancer risk $G(t)$ was compared with the cumulative risk of (a) esophageal cancer, (b) leukemia, (c) liver cancer, (d) lung cancer, (e) thyroid cancer, (f) pancreatic cancer, (g) colon cancer, (h) breast cancer, and (i) prostate cancer. Blue and red lines are the cases of males and females, respectively. Dots indicate cancer registry data of cumulative risk of cancer for the age range of 5–85. Curves are the fitting of $G(t)$ to data when $\lambda_2 = 0$. Fitted parameters are shown in each figure. The other parameters values are summarized in Table 1.

mutagens. However, as there are few cancers caused by errors, the maximum value of $RD$ becomes large. The influence of the number of cancer genes is contrary to the turnover rate. If carcinogenesis needs fewer mutations, the increase of $RD$ is faster, and the maximum value is smaller (Fig 5C). In this case, a relatively low number of mutations are sufficient for the complete mutation of all cancer driver genes; therefore, carcinogenesis occurs early because of

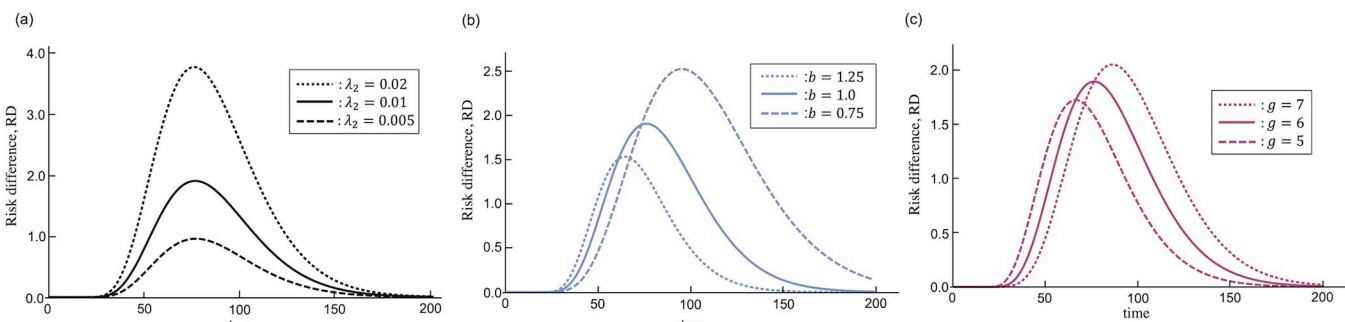

**Fig 5. Time course of risk difference with various parameters.** Parameter values are $N_0 = 1$, $N_L = 10^9$, $r = \log[N_L]/18$, $\lambda_1 = 0.5$, $p = 0.2$, (a) $b = 0.1$, $g = 6$, (b) $\lambda_2 = 0.01$, $g = 6$, and (c) $b = 1$, $\lambda_2 = 0.1$. The remaining parameters are shown in each figure.

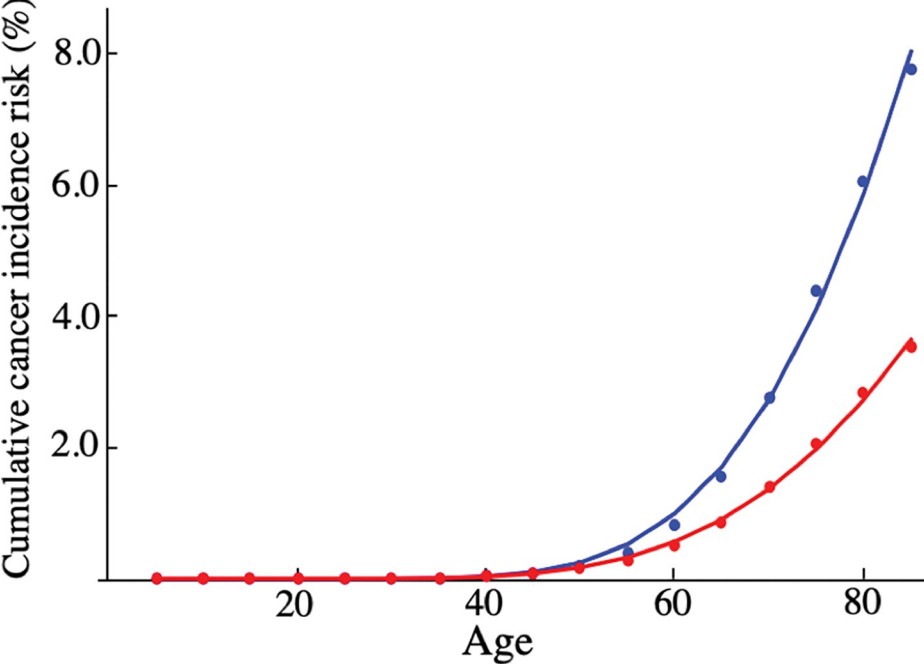

**Fig 6. Comparison of models with lesion and lung cancer data of age-dependent cancer risk.** Dots indicate cancer registry data of cumulative risk of cancer for the age range of 5–85. Curves are the fitting of $G(t)$ when $\lambda_2 = 0$ if the age is $< 20$, $\lambda_2$ is a constant value if the age is $> 20$. The other parameters values are summarized in Table 1.

additional mutations caused by mutagens. By contrast, the maximum value of RD is low because cancer arises relatively frequently without mutagens. These results suggest that if the turnover rate is low and/or the number of cancer driver gene mutations is large, the influence of mutagens emerges slower and larger. We re-analyzed the data on lung cancer to consider the influence of lesions (Fig 6). In this analysis, we assumed that lesions would accumulate after an age of 20 years. That is, $\lambda_2 = 0$ before age 20, and $\lambda_2$ has a constant value after that. This is because the smoking age in Japan is 20. As the fitting was already good even when considering replication errors only, it did not improve markedly. However, the estimated values were smaller than when considering replication errors only.

## Discussion

We constructed a mathematical model to discuss the risk of cancer caused by the accumulation of mutations. In the model, a single stem cell divides up to a certain number, $N_L$, and stops dividing as long as the total cell number can remain at $N_L$. Cell division resumes when cells are removed. Mutations are caused by errors and lesions. Errors arise stochastically with cell division, and lesions arise without cell division. First, we simulated the accumulation of errors and lesions using the Gillespie algorithm (Fig 2). The accumulation of errors and lesions were approximated and compared with the simulation (Fig 3). As so many cell divisions occur in

**Table 2. Estimated parameters from fitting and related parameters from previous studies.**

| | Estimated Number of cancer driver genes ($g$) | | Average number of cancer driver genes mutations per cell division. ($\lambda_1 p$) | | Number of driver mutations from previous studies | Number of cancer driver genes mutations per cell division estimated from [18–20] |
|---|---|---|---|---|---|---|
| | Male | Female | Male | Female | | |
| Leukemia | 2 | 2 | 0.0016 | 0.00014 | 1.94** | 0.000228, 0.000274 |
| Lung cancer | 10 | 7 | 1.045 | 0.54 | 5.33**, 5.16** | 0.000228, 0.000274 |
| Pancreatic cancer | 5 | 6 | 0.0226 | 0.0290 | 4.25* | 0.000228, 0.000274 |
| Colon cancer | 5 | 5 | 0.00041 | 0.00037 | 3.59*, 3.74** | 0.000228, 0.000274 |
| Breast cancer | - | 2 | - | 0.0018 | 3.32*, 1.76** | 0.000228, 0.000274 |

*Values from [3]

**Values from [17]

real tissue, calculation with the Gillespie algorithm takes a long time. Using these approximations, it is easier to compare with cancer registry data than using a simulation.

Next, we considered the probability of carcinogenesis based on the multistage model. Cancer risk is correlated to the number of stem-cell divisions [5]. This suggests that cancer risk might be explained by the accumulation of replication errors. In our model, this means that cancer risk is explained without lesions. We fitted our model to cancer registry data from the Cancer Information Service, National Cancer Center, Japan (Fig 4). Except for leukemia, our model fitted to the risks of cancers. This supports the hypothesis that the risks of some cancers are explained by replication errors. Leukemia was not explained by our model as the risk occurs at a very young age. The model assumes that cells divide vigorously at a younger age. If we try to explain the early risk of leukemia through replication errors, the risk in the stable phase becomes too large because hematopoietic stem cells are considered to divide even in adults. It is said that chronic myeloid leukemia can be explained by a single mutation when the mutated cells acquire reproductive advantages [4]. We assumed that mutations do not affect the biological characters of the cells in this study. Therefore, the influence of mutations on cells should be considered in future studies. We estimated 3–10 cancer driver gene mutations. Some studies assessed the numbers of cancer-related mutations (Table 2).

Vogelstein et al. [3] reviewed the distribution of the numbers of driver mutations in tumors. Pancreatic cancer, colorectal cancer, and breast cancer were associated with, on average, 4.25, 3.59, and 3.32 mutations, respectively. A different study analyzed 3,281 tumors from 12 cancer types and identified 127 genes that showed significant frequencies of mutations [17]. The average numbers of mutations in colon and rectal carcinoma, acute myeloid leukemia, lung adenocarcinoma, lung squamous cell carcinoma, and breast adenocarcinoma were 3.74, 1.94, 5.33, 5.16, and 1.76, respectively. The estimation of the present study showed that leukemia required 2 mutations, pancreatic cancer requires 5 or 6 mutations, colon cancer requires 5 mutations, and breast cancer requires 2 mutations. These estimations therefore deviate only slightly from the previously reported values. However, the estimated value of $g$ = 10 or 7 for lung cancer is larger than the values. In addition, the estimated mutation rate $\lambda_1 p$ of lung cancer is very large. The mutation rate is estimated at approximately 1.14 mutations per genome per division in healthy haematopoiesis and 1.37 per genome per division in brain development [18], which

corresponds to $\lambda_1$. Then, $p$ can be interpreted as the proportion of cancer genes in the genome. Combined with the estimation that more than 1% of the genes contribute to human cancer [19] and the estimation that the protein-coding gene is 2% [20], $p$ is considered to be approximately 0.01×0.02 = 0.0002. From these estimations, $\lambda_1 p$ will be approximately 1.14×0.0002 = 0.000228 or 1.37×0.0002 = 0.000274. The estimated values of $\lambda_1 p$ are too large in liver, lung, thyroid, pancreatic, breast, and colon cancer. This may suggest that biological processes other than replication errors affect the cancer risk. We discuss the biological processes that can affect the results in the last section.

The change in cancer risk caused by mutagens is an important problem in the field of oncology. For example, high dose and dose-rate radiation increase cancer risk [9]. However, the influence of very low dose and dose-rate radiation, which is important for radiation protection, is unclear [11]. We discussed the influence of lesions caused by mutagens on cancer risk using risk difference (*RD*) as an indicator (Fig 5). *RD* increased initially and then began to decrease. The decrease in the latter half is due to the assumption that carcinogenesis occurs even in the absence of mutagens because of the accumulation of replication errors. In reality, the decrease will be difficult to observe because life is limited. The timing of the start of *RD* increases depending on biological parameters. The increase is delayed when the cell-removal rate $b$ is low (Fig 5B) and/or the number of cancer driver genes $g$ is large (Fig 5C). The removal rate $b$ can be regarded as the division rate if the removal of a stem cell is immediately compensated by the cell division of the remnant stem cells. For example, astrocytes and neural stem cells are the origins of glioblastoma and medulloblastoma, respectively. The number of divisions of those stem cell per year is estimated at 0 [5]. The influence of mutagens on these cancer risks might appear slowly. However, it is estimated that colonic stem cells divide 75-times per year [5]. The influence of mutagens might be observed relatively early in the colon. Recently, a hypothesis that radiation, especially high dose-rate radiation, might lead to an earlier onset of cancer has been advocated [21]. The time course of RD in this study focuses on cancer risk change with time. When risk $G$ is fixed and solved with respect to $t$, the difference might be an indicator of the early onset of cancer.

In order to integrate the effects of mutagens, we focused on lung cancer, which showed a large difference between estimated parameters when only replication errors and previously reported values. In addition, the estimated the number of cancer driver genes in lung cancer also largely differed between sexes. This discrepancy will be partly due to incompleteness of the estimation, as discussed below, but can be partially resolved by assuming mutagen. It is known that smoking increases lung cancer risk (e.g., [22]). The data included smokers and non-smokers, with more male than female smokers. This may lead to a larger estimated value for the number of cancer driver genes in male data than in female data and the large difference between sexes. When we integrated the influence of mutagen, the fitting did not differ as much. However, the estimated value was smaller and closer to the previously reported values than in the case of replication error alone. The difference between sexes became slightly smaller. By considering the effects of mutagens in more detail, the effects of replication errors might be estimated more accurately.

The present analysis was very simple, and there are limitations that must be solved in the future to derive more accurate results. We did not consider population fluctuation when fitting the cancer registry data. In addition, fitting by the least-squares method was performed, but this assumes an age-independent normal distribution. These may have caused the differences between sexes in the parameter estimates. For more accurate estimates, it will be necessary to take into account demographics and different health conditions among generations. Moreover, this study ignored many biological processes. The end of growth phase was assumed at age 18, but the actual age may vary depending on the organ or tissue. Assuming a longer

growth phase, estimated $g$ and $\lambda_1 p$ would be smaller. We assumed that properties of the cells were independent of errors and lesions. However, the acquisition of driver mutations may increase the growth rate, and mutations in DNA damage repair genes may increase the mutation rate. In addition, as the accumulation of mutations was treated as the average of the population in the multi-stage carcinogenesis model, polyclonality of cancer could not be expressed. Many cancers are known to be polyclonal [23]. It may be possible to incorporate changes in cell properties and cancer polyclonality by focusing on each cell, rather than on the population average, in the process of carcinogenesis. For example, if the properties of each cell are changed by mutations, $a_{\{i,j\}}$, $b_{\{i,j\}}$, and $\lambda_1$ should depend on the number of errors $i$ and that of lesions $j$. In the stable phase, the division rate was constant; however, it is known to decreases with age [7]. It is also known that the effect of radiation is more pronounced in younger people than in adults [9]. The relationship between division rates and mutagens will also be an important issue when considering radiation protection. For example, assuming that $a_{\{i,j\}}$ and/or $\lambda_2$ are time-dependent, it may be possible to incorporate those phenomena. Moreover, we assumed that the interaction of stem cells does not affect the proliferation and removal of cells. However, cells exposed to mutagens such as ionizing radiations might be removed through stem-cell competition [24, 25]. If so, the removal rate of damaged cells may be larger than that of intact cells. These phenomena should be considered in the future to examine cancer risk in more detail.

## Supporting information

**S1 Fig. Estimation of $g$ and $\lambda_1 p$ through the least squares method.** The sum of the squares of the residuals for each $g$ was minimized by $\lambda_1 p$. Then, a combination of $g$ and $\lambda_1 p$ showing the minimum of the sum of squares of the residuals was established. The order of the figures corresponds to that in Fig 5: (a) esophageal cancer, (b) leukemia, (c) liver cancer, (d) lung cancer, (e) thyroid cancer, (f) pancreatic cancer, (g) colon cancer, (h) breast cancer, and (i) prostate cancer. The upper and lower panels show the results of males and females, respectively. (TIF)

**S1 File. Code of Gillespie algorithm.** This is a C language code for Gillespie algorithm. The default parameters correspond to Fig 2. (C)

## Acknowledgments

We thank Dr. Hiroshi Haeno for technical discussions on the mathematical model.

## Author Contributions

**Conceptualization:** Kouki Uchinomiya, Masanori Tomita.

**Data curation:** Kouki Uchinomiya, Masanori Tomita.

**Formal analysis:** Kouki Uchinomiya.

**Funding acquisition:** Kouki Uchinomiya, Masanori Tomita.

**Investigation:** Kouki Uchinomiya, Masanori Tomita.

**Methodology:** Kouki Uchinomiya, Masanori Tomita.

**Project administration:** Kouki Uchinomiya, Masanori Tomita.

**Resources:** Kouki Uchinomiya, Masanori Tomita.

**Software:** Kouki Uchinomiya.

**Supervision:** Masanori Tomita.

**Validation:** Kouki Uchinomiya, Masanori Tomita.

**Visualization:** Kouki Uchinomiya.

**Writing – original draft:** Kouki Uchinomiya, Masanori Tomita.

**Writing – review & editing:** Kouki Uchinomiya, Masanori Tomita.

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
