## [Decision Letter · Decision Letter 0]

30 May 2022

PONE-D-22-12989A mathematical model for cancer risk and accumulation of mutations caused by replication errors and external factorsPLOS ONE

Dear Dr. Uchinomiya,

Thank you for submitting your manuscript to PLOS ONE. After careful consideration, we feel that it has merit but does not fully meet PLOS ONE’s publication criteria as it currently stands. Therefore, we invite you to submit a revised version of the manuscript that addresses the points raised during the review process.One reviewer suggested major revisions, the other recommended rejection.  Please attempt to address as many of the comments as possible.

We look forward to receiving your revised manuscript.

Kind regards,

Gayle E. Woloschak, PhD

Section Editor

PLOS ONE

Journal Requirements:

 "KU was supported by JSPS KAKENHI Grant Number JP 20K19972."  

Additional Editor Comments :

One reviewer recommended rejection, the other asked for major revisions. Please address as of these concerns as possible in a revision.

Reviewers' comments:

Reviewer's Responses to Questions

**Comments to the Author**

1. Is the manuscript technically sound, and do the data support the conclusions?

Reviewer #1: No

Reviewer #2: Partly

2. Has the statistical analysis been performed appropriately and rigorously? 

Reviewer #1: I Don't Know

Reviewer #2: Yes

3. Have the authors made all data underlying the findings in their manuscript fully available?

Reviewer #1: Yes

Reviewer #2: Yes

4. Is the manuscript presented in an intelligible fashion and written in standard English?

Reviewer #1: Yes

Reviewer #2: Yes

5. Review Comments to the Author

Reviewer #1: The described exercise investigates the multistage model, adding complexity in terms of including developing and stable tissue compartments. Overall the modeling lacks biological significance (i.e., no distinction regarding functions caused by mutations, assumes monoclonal tumor origin, assumes mutations rates that were not justified. Consequently, the value of the exercise is questionable. Also, identifying exogenously-induced mutations as wounds is non-standard nomenclature.

Reviewer #2: Uchinomiya and Tomita present a mathematical model for accumulation of mutations during development and in adult tissues. They use it to fit cancer risk data, inferring that accumulation of DNA copying-errors is largely sufficient to explain cancer risk across many cancer types. Overall the study would appear sound, if the issues described below (1-3) are addressed. There is considerable interest in deriving these sorts of models for cancer risk and I think such studies could benefit from incorporating some cancer genomic data that is now available.

Major:

1- One major point of concern is that for the central result -- the fits to epidemiological cancer data (as shown in Figure 3) -- the inferred parameters "lambda_1*p" (number of oncogenic driver mutations acquired per cell division) and "g" (number of driver mutations required) for each cancer type were largely not constrained to match biological knowledge of each cancer type, nor were they checked to see if the inferred values match what we expect.

There are some suspect examples where the "lambda_1*p" parameter can vary by >3 orders of magnitude between e.g. panel (g) [colon] and panel (d) [lung]. I think it is very unlikely tht the number of errors during cell division could vary so much between different tissues. The number of mutations per division has been measured recently by WGS for a variety of healthy human tissues; this data should be looked up and compared to the estimates of this parameter, to check veracity.

1b- A related point is the number of driver mutations in the "g" parameter: that this would be 7-10 for lung cancer, 4-5 for esophagus, while 2 for thyroid and breast is a bit unusual. Does this match the actually observed number of drivers in these tissues in e.g. TCGA or similar cancer genomics data sets? (I appreciate it is not trivial to exactly measure the number of driver mutations per cancer type however some estimates can be made). If this doesn't match the models should be ajusted.

2- There is a worry about whether multiple solutions with different combinations of their 2 parameters "lambda_1*p" (number of oncogenic driver mutations acquired per cell division) and "g" (number of driver mutations required) would fit the cancer incidence data similarly well, and thus might also be plausible solutions. Thus the fit might be visualized as a heatmap of the possible solutions. This fits the biological intuition that e.g. the somatic mutation rate will be somewhat different across individuals (they might have a germline defect in a DNA repair gene or not), and also that the number of driver mutations is variable within a cancer type -- it is a range, and not (as now somewhat artificially contstrained) a single value.

3- I think related with the above: there should be a formal test (perhaps based on bootstrap, or similar) to test if the addition of the lambda_2 parameter (mutagen exposure/wounds) significantly increases the fit of the models to the epidemiological data, or not. Now there is a statement about how the mutagen exposure factor is not necessary in the models to achieve a good fit to the cancer risk curves, however this doesn't seem to be supported with actual data. Does the fit improve with adding this (even if the improvement is subtle)?

Needs dissussing:

- In at least some adult tissues, stem cells are not renewed during aging at the rate at which they are lost; their number declines with age. E.g. blood (HSC) is one example and there may be others. My intution is this should not impact their model massively. Can they comment/discuss, or reanalyze if necessary?

- A tricky problem is that mutations resulting from DNA lesions ("wounds") might be occuring in a way coupled with DNA replication. Replication may increase the rate of converting wounds into mutations. If cells replicate slowly, there is plenty of time for repair e.g. via the NER pathway. If they replicate fast, lesions are not repaired but instead copied-over using TLS enyzmes thus generating mutations. So an ideal model would also account for this interaction between replication rate/ mutations and 'wound' exposures, having this as a parameter. I am not sure if such (more complicated) parametrization is feasible here, given the few amount of data points to fit to. Can they discuss this and/or adress with analysis if they think important? See the example of the "lambda_1*p" parameter (allegedly replication-dependant but not mutagen exposure dependant) being higher in lung; this is odd.

- One idea for future work, which I understand might be out-of-scope here, is to model explicitly the existence of subsets of mutator cancers (e.g. MSI colon cancers; the BRCA breast cancer).

Minor:

- The mathematical formalism may be hard to follow for some readers without an appropriate background. Text describing the method is well described however takes effort to absorb. I suggest having a visual diagram/schematic describing the simulation method.

- In figure 3, write cancer type on the figure panels. In Fig 2, write color legend in the figure.

- Wording "Wound", while not incorrect, suggests a macroscopic event. In DNA, the word "lesion" is more commonly used.

- I don't understand much the emphasis on ionizing radiation in the abstract, while there are no analyses particular to this in the manuscript. It is also probably not a major carcinogen compared to e.g. tobacco smoking.

6. PLOS authors have the option to publish the peer review history of their article (what does this mean?). If published, this will include your full peer review and any attached files.

Reviewer #1: No

Reviewer #2: No

---

## [Author Response · Author response to Decision Letter 0]

21 Jul 2022

>Reviewer #1: The described exercise investigates the multistage model, adding complexity in

>terms of including developing and stable tissue compartments. Overall the modeling lacks 

>biological significance (i.e., no distinction regarding functions caused by mutations, assumes

>monoclonal tumor origin, assumes mutations rates that were not justified. Consequently, the 

>value of the exercise is questionable. Also, identifying exogenously-induced mutations as 

>wounds is non-standard nomenclature.

We appreciate the comments of this reviewer #1. We understand the concerns of this reviewer and revised the manuscript. 

[Answers] 

As reviewer #1 pointed out, this model ignores many biological processes. The goal of this paper is to examine how the cancer risks can be explained by biological processes as simple as possible. We couldn't express this point well, so we added descriptions (Line 82-85).

Although we ignored many biological processes, it is not difficult to incorporate some of the neglected processes. The model can include functions caused by mutations through parameters a_{i,j} and b_{i,j}(We added descriptions in Line 493-496, 500-502). However, it is very difficult to assume how mutations affect realistically. Therefore, for convenience, a_{i,j} and b_{i,j} were assumed to be constant. As the descriptions about this simplification were lacking, we added the explanation (Line 201-203).

As with the classic model such as Armitage & Doll (1954), this model also assumes a monoclonal tumor origin. This model is a monoclonal tumor origin because it assumes the population average in the carcinogenesis process. If we focus on individual cells, polyclonality can be expressed. We added a discussion about this (Line 496-500).

We added references and discussions on mutation rate (Line 449-459) and changed the term “wound” to “lesion”.

>Reviewer #2: Uchinomiya and Tomita present a mathematical model for accumulation of 

>mutations during development and in adult tissues. They use it to fit cancer risk data, inferring 

>that accumulation of DNA copying-errors is largely sufficient to explain cancer risk across 

>many cancer types. Overall the study would appear sound, if the issues described below (1-3) 

>are addressed. There is considerable interest in deriving these sorts of models for cancer risk 

>and I think such studies could benefit from incorporating some cancer genomic data that is 

>now available.

We appreciate the positive evaluation of our work by reviewer #2. We addressed all suggestions.

>Major:

>1- One major point of concern is that for the central result -- the fits to epidemiological cancer 

>data (as shown in Figure 3) -- the inferred parameters "lambda_1*p" (number of oncogenic 

>driver mutations acquired per cell division) and "g" (number of driver mutations required) for 

>each cancer type were largely not constrained to match biological knowledge of each cancer 

>type, nor were they checked to see if the inferred values match what we expect.

>There are some suspect examples where the "lambda_1*p" parameter can vary by >3 orders of 

>magnitude between e.g. panel (g) [colon] and panel (d) [lung]. I think it is very unlikely tht the 

>number of errors during cell division could vary so much between different tissues. The 

>number of mutations per division has been measured recently by WGS for a variety of healthy 

>human tissues; this data should be looked up and compared to the estimates of this 

>parameter, to check veracity.

[Answer]

The mutation rate is estimated approximately 1.14 mutations per genome per division in healthy haematopoiesis and 1.37 per genome per division in brain development (Warner et al. 2020), which corresponds to lambda_1. Then, p can be interpreted as the proportion of cancer genes in the genome. Combined with the estimation that more than 1% of genes contribute to human cancer (Futreal et al. 2004) and the estimation that the protein-coding gene is 2% (IHGSC, 2004), lambda_1 * p is considered to be about 0.000228 or 0.000274. The values are too large in liver, lung, thyroid, pancreatic, breast, and colon cancer. This may suggest that biological processes other than replication errors affect the cancer risk. We added descriptions and discussions (Line 449-459). 

>1b- A related point is the number of driver mutations in the "g" parameter: that this would be 

>7-10 for lung cancer, 4-5 for esophagus, while 2 for thyroid and breast is a bit unusual. Does 

>this match the actually observed number of drivers in these tissues in e.g. TCGA or similar 

>cancer genomics data sets? (I appreciate it is not trivial to exactly measure the number of 

>driver mutations per cancer type however some estimates can be made). If this doesn't match 

>the models should be ajusted.

[Answer]

We referred to some papers for discussing the number of driver mutations. The parameter “g” of some cancers was not so different from the observation, but the others were different. Ignoring the biological processes in this paper may cause this mismatch, so we discussed it (Line 436-448).

>2- There is a worry about whether multiple solutions with different combinations of their 2 

>parameters "lambda_1*p" (number of oncogenic driver mutations acquired per cell division) 

>and "g" (number of driver mutations required) would fit the cancer incidence data similarly 

>well, and thus might also be plausible solutions. Thus the fit might be visualized as a heatmap 

>of the possible solutions. This fits the biological intuition that e.g. the somatic mutation rate 

>will be somewhat different across individuals (they might have a germline defect in a DNA 

>repair gene or not), and also that the number of driver mutations is variable within a cancer 

>type -- it is a range, and not (as now somewhat artificially contstrained) a single value.

[Answer]

We agree with the concern that different combinations of the 2 parameters may be the solution. Although we tried to show it by heatmap, it were hard to see. Alternatively, we showed the relationship among residual sum of squares and the 2 parameters for each fitting (S Fig1). 

>3- I think related with the above: there should be a formal test (perhaps based on bootstrap, or 

>similar) to test if the addition of the lambda_2 parameter (mutagen exposure/wounds) 

>significantly increases the fit of the models to the epidemiological data, or not. Now there is a 

>statement about how the mutagen exposure factor is not necessary in the models to achieve a 

>good fit to the cancer risk curves, however this doesn't seem to be supported with actual data. 

>Does the fit improve with adding this (even if the improvement is subtle)?

[Answer]

The effect of the mutagen will be best understood in lung cancer, but since the fitting itself is successful even with the error, considering the effect of the mutagen may not improve the fitting. However, when considering only the duplication error, the estimated parameters are significantly different from the values known in experiments. When considering the mutagen, the parameters are close to the observed values to some extent. We added a figure and a discussion about this (Fig 6, Line 391-397, Line 482-491).

>Needs dissussing:

>- In at least some adult tissues, stem cells are not renewed during aging at the rate at which 

>they are lost; their number declines with age. E.g. blood (HSC) is one example and there may 

>be others. My intution is this should not impact their model massively. Can they 

>comment/discuss, or reanalyze if necessary?

[Answer]

When the division rate decreases, the accumulation of replication errors in old age is overestimated in this model. It is necessary to consider the change in the division rate when constructing a more accurate model. We will be able to incorporate the effect by changing the division rate a_ {i, j} depending on time. We added a discussion about this (Line 502-507).

>- A tricky problem is that mutations resulting from DNA lesions ("wounds") might be occuring 

>in a way coupled with DNA replication. Replication may increase the rate of converting wounds 

>into mutations. If cells replicate slowly, there is plenty of time for repair e.g. via the NER 

>pathway. If they replicate fast, lesions are not repaired but instead copied-over using TLS 

>enyzmes thus generating mutations. So an ideal model would also account for this interaction 

>between replication rate/ mutations and 'wound' exposures, having this as a parameter. I am 

>not sure if such (more complicated) parametrization is feasible here, given the few amount of 

>data points to fit to. Can they discuss this and/or adress with analysis if they think important? 

>See the example of the "lambda_1*p" parameter (allegedly replication-dependant but not 

>mutagen exposure dependant) being higher in lung; this is odd.

[Answer]

We are interested in the relationship between replication rate and lesions (we changed the term from “wounds”). The influence of radiation exposure on younger people is known to increase the cancer risk compared to adults. The relationship between replication rate and accumulation of lesions may also be important in discussing radiation effects. In terms of this model, it is conceivable to change lambda_2 with age or replication rate. We added discussions about this while addressing the comment above. (Line 503-507).

>- One idea for future work, which I understand might be out-of-scope here, is to model 

>explicitly the existence of subsets of mutator cancers (e.g. MSI colon cancers; the BRCA breast 

>cancer).

[Answer]

Indeed, this model only considers the average properties of cells in each tissue. In mutator cancers, lambda_1 related to the mutation rate will be larger in other cancers. I added a discussion about it (Line 459-467)

>Minor:

>- The mathematical formalism may be hard to follow for some readers without an appropriate 

>background. Text describing the method is well described however takes effort to absorb. I 

>suggest having a visual diagram/schematic describing the simulation method.

[Answer]

We added a schematic diagram as Fig1 and revised the numbers in other figures.

>- In figure 3, write cancer type on the figure panels. In Fig 2, write color legend in the figure.

[Answer]

We added a color legend in Fig 2(new number is Fig 3) and cancer type on the figure panels in Fig 3(new number is Fig 4). 

>- Wording "Wound", while not incorrect, suggests a macroscopic event. In DNA, the word 

>"lesion" is more commonly used.

[Answer]

We changed the term “wound” to “lesion”.

>- I don't understand much the emphasis on ionizing radiation in the abstract, while there are 

>no analyses particular to this in the manuscript. It is also probably not a major carcinogen 

>compared to e.g. tobacco smoking.

[Answer]

Our aim is to create a model for discussing the effects of low-dose and low-dose-rate radiation, which is difficult to observe. We emphasized that the model is effective for analyzing such phenomena. (L82-85)

---

## [Decision Letter · Decision Letter 1]

31 Oct 2022

PONE-D-22-12989R1A mathematical model for cancer risk and accumulation of mutations caused by replication errors and external factorsPLOS ONE

Dear Dr. Uchinomiya:

Thank you for submitting your manuscript to PLOS ONE. After careful consideration, we feel that it has merit but does not fully meet PLOS ONE’s publication criteria as it currently stands. Therefore, we invite you to submit a revised version of the manuscript that addresses the points raised during the review process.

Major revisions have been recommended by the reviewers.  Please address concerns in a revision.

We look forward to receiving your revised manuscript.

Kind regards,

Gayle E. Woloschak, PhD

Section Editor

PLOS ONE

Additional Editor Comments (if provided):

One reviewer suggested major revisions, one suggested minor revisions. Please address these concerns in your revised manuscript.

Reviewers' comments:

Reviewer's Responses to Questions

**Comments to the Author**

1. If the authors have adequately addressed your comments raised in a previous round of review and you feel that this manuscript is now acceptable for publication, you may indicate that here to bypass the “Comments to the Author” section, enter your conflict of interest statement in the “Confidential to Editor” section, and submit your "Accept" recommendation.

Reviewer #2: All comments have been addressed

Reviewer #3: (No Response)

2. Is the manuscript technically sound, and do the data support the conclusions?

Reviewer #2: Yes

Reviewer #3: Partly

3. Has the statistical analysis been performed appropriately and rigorously? 

Reviewer #2: Yes

Reviewer #3: Yes

4. Have the authors made all data underlying the findings in their manuscript fully available?

Reviewer #2: Yes

Reviewer #3: No

5. Is the manuscript presented in an intelligible fashion and written in standard English?

Reviewer #2: Yes

Reviewer #3: Yes

6. Review Comments to the Author

Reviewer #2: The authors have adequately adressed my concerns with additional discussion.

It is especially appreciated that they explicitly comment on the cases where their parameter fits are very different from realistic values, which may reveal interesting biology: "The estimated values of 1 are too large in liver, lung, thyroid, pancreatic, breast, and colon cancer. This may suggest that biological processes other than replication errors affect the cancer risk." One final request I would have that this result " 1 are too large in liver, lung, thyroid, pancreatic, breast, and colon cancer", which is in my opinion important (not to invalidate the model but rather to suggest effects of additional biological factors), be shown in a separate figure or table. In other words, I would suggest to include a figure/table that compares the fitted parameters to a (range of) expected parameters from the literature, for each cancer type.

As a minor comment, the sentence in the abstract "The parameters estimated by the analysis of lung cancer data were closer to the observed values than when considering only replication errors." is quite unclear as written; consider expanding it to clearly convey what is the meaning/implication of this lung result.

Finally it it still unclear to me why the focus on radiation in the introduction if there is not any radiation specific analysis. It is not wrong, but is a bit confusing perhaps.

Reviewer #3: The present reviewer has been added in the revision process. The study is well designed and addresses an interesting topic. However, there are several unclarities which need to be solved before acceptance for publication.

Major comments

1. Classically, as in ref. 2, the Armitage-Doll model is fitted to the rate (i.e., cases per person-year) unlike the present study, where the cumulative rate data (i.e., cases per total population) is used for fitting (equations 24 and 25). Please comment whether the “cumulative rate” data of the cancer registry herein are adjusted for competing factors (such as the age-related reduction in population). In addition, the cumulative rate at a certain age is dependent on the rates at all earlier time points because of its cumulative nature. This indicates that the uncertainty of the data is also accumulated with age. Fitting using the least square method assumes identical (i.e., age-independent) normal distribution on the uncertainty, and thus, does not take into account this cumulative nature of the uncertainty. Please consider stating these points, if applicable, as limitations of the study.

2. In Fig. 2, the increase of N(t) in the growth phase seems to slow when it gets close to 300. This is not expected from the description of the model. If the authors modeled some slowing in the proliferation rate when the stem cell pool is nearly full, please comment on this in the Method section. If not, please explain the mechanism of the slowing.

3. Equation 29 may need reconsideration. Because lambda_2 denotes the rate at which W changes in a cell, a differential equation of “dW/dt = lambda_2 * N_0 * exp[rt]” should hold. Solving this under an assumption of W(0) = 0, one obtains the equation of “W = (lambda_2 * N_0 / r) * (exp[rt] – 1)” instead of equation 29.

4. p.17 line 275 and equation 32. I do not understand why “a – b” is used here instead of “a”. Please consider using “a” or explaining why “a – b” is appropriate. I understand that the assumption described in lines 273-274 should be dealt with by the assumption of “a >> b” rather than the use of “a – b”.

5. In equation 40, the authors assume that exponential growth of the stem cell pool continues until age 18, which is quite counterintuitive, while this assumption contributes to the simplification of the model. Please discuss the consequence of changing the age at which the growth phase is terminated.

6. p.20 line 326 “lambda_1 * p is regarded as a single parameter in Eq (25)” This statement may not be obvious to readers because “lambda_1 * p” is not explicit in Eq (25). Please consider showing the relevant calculation process.

7. Fig. 4 and S1 Fig. indicate different values of g between the sexes, which may seem odd. Please consider using an identical value for both sexes or showing a rationale for using different values (such as previous observations supporting a sex-dependent number of driver gene mutations).

8. p.29 lines 472-474. Ovarian germ cells may not be the only cell-of-origin of ovarian cancer because somatic cells of the ovary and the oviduct can also be its origin. Please reconsider the context.

9. Please consider code sharing as stated in https://journals.plos.org/plosone/s/materials-software-and-code-sharing.

Minor points

p.3 lines 31-32 and other places: Please consider adding the word “previous” when referring to previous observations. Example: … the estimated parameters did not always agree with “previously reported values”. (p.3 lines 31-32)

p.47 line 47: “appearance” should be “mortality” according to ref. 2.

p.6 lines 74-75: Consider the following modification: “… and ionizing radiation contributes to the carcinogenic process by adding a few mutations”

p.6 line 91: … due “to” replication errors …

p.7 line 106, p.24 line 387, p.31 line 503: Replace the first comma (,) with a semicolon (;) (e.g., …; however, …).

p.12 lines 184-185: “…, in the growth phase, …” This phrase should be deleted.

p.12 line 186: The subscript g should be substituted with s (Gamma_g to Gamma_s).

p.13 line 207 and other occasions: Because “oncogenes” do not include “tumor suppressor genes”, the wording here should be “cancer driver genes” instead of “oncogenes”. The same applies to many occasions in the manuscript.

p.14 line 217: “… of having more than g mutations …” should be “… of having g or more mutations …”

p.14 line 221: “cancer risk” should be “cumulative cancer incidence”

p.16 line 245 and p.19 line 305: The value of N_L seems to be 300 instead of 100, as the maximum value of N in Fig 2 is 300.

p.16 line 252: n should be d?

p.17 line 273: the other -> another

p.19 line 308 and other occasions: The wording of “statistical data” is odd and should be replaced with expressions like “epidemiology data”, “real-world data” or “cancer registry data”.

p.20 line 314: The ministry name should be followed by the country name.

p.20 line 328: “value” -> “values”

p.24 line 385: “larger” -> “smaller”?

p.29 line 480: “with on time” -> “with time”?

p.30 lines 501-502: The grammar of “… should be depends on …” needs reconsideration.

7. PLOS authors have the option to publish the peer review history of their article (what does this mean?). If published, this will include your full peer review and any attached files.

Reviewer #2: No

Reviewer #3: No

---

## [Author Response · Author response to Decision Letter 1]

3 Feb 2023

>Reviewer #2: The authors have adequately adressed my concerns with additional discussion.

[Answer] We were relieved that our response seemed to address your concerns. We have responded to all Reviewer #2’s comments.

>It is especially appreciated that they explicitly comment on the cases where their parameter fits are 

>very different from realistic values, which may reveal interesting biology: "The estimated values of 

>1 are too large in liver, lung, thyroid, pancreatic, breast, and colon cancer. This may suggest that 

>biological processes other than replication errors affect the cancer risk." One final request I would 

>have that this result " 1 are too large in liver, lung, thyroid, pancreatic, breast, and colon cancer", 

>which is in my opinion important (not to invalidate the model but rather to suggest effects of 

>additional biological factors), be shown in a separate figure or table. In other words, I would suggest 

>to include a figure/table that compares the fitted parameters to a (range of) expected parameters 

>from the literature, for each cancer type.

[Answer]We added the table comparing fitted parameters and expected parameters (Table 2). 

>As a minor comment, the sentence in the abstract "The parameters estimated by the analysis of lung 

>cancer data were closer to the observed values than when considering only replication errors." is 

>quite unclear as written; consider expanding it to clearly convey what is the meaning/implication 

>of this lung result.

[Answer]

We reconsidered the sentence (Line 33-34, 41-43). 

>Finally it still unclear to me why the focus on radiation in the introduction if there is not any 

>radiation specific analysis. It is not wrong, but is a bit confusing perhaps.

[Answer]

As we added in line 83-85, the carcinogenic effects of low-dose and low-dose-rate radiation have become a major public concern in Japan since the accident at the Fukushima Daiichi Nuclear Power Plant. The results of this paper are expected to make a significant contribution to the evaluation of cancer risk from low-dose and low-dose-rate radiation. Therefore, we would like to keep descriptions about radiation.

>Reviewer #3: The present reviewer has been added in the revision process. The study is well 

>designed and addresses an interesting topic. However, there are several unclarities which need to 

>be solved before acceptance for publication.

[Answer] 

Thank you for your important comments. We have carefully considered and responded to those comments.

>Major comments

>1. Classically, as in ref. 2, the Armitage-Doll model is fitted to the rate (i.e., cases per person-year) 

>unlike the present study, where the cumulative rate data (i.e., cases per total population) is used for 

>fitting (equations 24 and 25). Please comment whether the “cumulative rate” data of the cancer 

>registry herein are adjusted for competing factors (such as the age-related reduction in population). 

>In addition, the cumulative rate at a certain age is dependent on the rates at all earlier time points 

>because of its cumulative nature. This indicates that the uncertainty of the data is also accumulated 

>with age. Fitting using the least square method assumes identical (i.e., age-independent) normal 

>distribution on the uncertainty, and thus, does not take into account this cumulative nature of the 

>uncertainty. Please consider stating these points, if applicable, as limitations of the study.

[Answer] 

As Reviewer #3 pointed out, there are limitations related to demographics and the appropriateness of using the least-squares method. We added the discussion about these limitations (Line 519-528).

>2. In Fig. 2, the increase of N(t) in the growth phase seems to slow when it gets close to 300. This is 

>not expected from the description of the model. If the authors modeled some slowing in the 

>proliferation rate when the stem cell pool is nearly full, please comment on this in the Method 

>section. If not, please explain the mechanism of the slowing.

[Answer] 

Since the results are output when a certain number of divisions occurred for reducing the drawing cost, it looks that the rate of increase slowing around N=300 due to smoothing in combining them. We clarified in the caption that we are connecting discrete values. In addition, we did a new simulation and redrew the figure. In the new simulation, we set N=100 which used in figure 3. 

>3. Equation 29 may need reconsideration. Because lambda_2 denotes the rate at which W changes 

>in a cell, a differential equation of “dW/dt = lambda_2 * N_0 * exp[rt]” should hold. Solving this under 

>an assumption of W(0) = 0, one obtains the equation of “W = (lambda_2 * N_0 / r) * (exp[rt] – 1)” 

>instead of equation 29.

[Answer] 

Thank you for your very helpful comments. We changed the approximation of W in the growth phase following the comments (eqs. 29 and 31). Following this change, the approximation of W in the stable phase must be changed, which was also addressed (eqs. 39 and 40). In addition, figures 3 and 5 are affected with this change and the Major-comment 4. We recalculated and redrew these figures.

>4. p.17 line 275 and equation 32. I do not understand why “a – b” is used here instead of “a”. Please 

>consider using “a” or explaining why “a – b” is appropriate. I understand that the assumption 

>described in lines 273-274 should be dealt with by the assumption of “a >> b” rather than the use of 

>“a – b”.

[Answer] 

We agree with this comment. Using “a” is more appropriate. We rewrote the related equations (eqs. 32-34, 36-38, 40). In addition, figures 3 and 5 are redrawn as mentioned in the response to the Major-comment 3

>5. In equation 40, the authors assume that exponential growth of the stem cell pool continues until 

>age 18, which is quite counterintuitive, while this assumption contributes to the simplification of 

>the model. Please discuss the consequence of changing the age at which the growth phase is 

>terminated.

[Answer] 

If the growth phase ends later, estimated g and lambda_1*p can be smaller. When the growth phase is long, the stable phase will be short, so it is necessary to develop cancer in a shorter time. Therefore, g should be small. Under this condition, cancer cells emerge earlier if lambda_1*p remains unchanged. To prevent this lambda_1*p should be smaller. I added a brief discussion about this (Line 526-527).

>6. p.20 line 326 “lambda_1 * p is regarded as a single parameter in Eq (25)” This statement may not 

>be obvious to readers because “lambda_1 * p” is not explicit in Eq (25). Please consider showing the 

>relevant calculation process.

[Answer] 

We can show this by combining eqs (25) and (37), so we added the explanation (Line 343-345). 

>7. Fig. 4 and S1 Fig. indicate different values of g between the sexes, which may seem odd. Please 

>consider using an identical value for both sexes or showing a rationale for using different values 

>(such as previous observations supporting a sex-dependent number of driver gene mutations).

[Answer] 

We believe the value of g should be the same for both sexes. We think that there are two factors that cause the differences in the estimation. One is that the fitting is incomplete as pointed out in Major-comments 1. Esophageal cancers and pancreatic cancers with small differences in g may correspond to this case. Another cause will be the limitation of considering cumulative cancer risk based only on replication errors. Estimated g of Liver cancer and Lung cancer differ more between sexes. In Lung cancer, the influence of smoking is well known as risk factor, and the difference in g between sexes became smaller when the influence of mutagen was assumed. We added it to the Discussion for clarifying about this (Line 507-510, 522-523).

>8. p.29 lines 472-474. Ovarian germ cells may not be the only cell-of-origin of ovarian cancer 

>because somatic cells of the ovary and the oviduct can also be its origin. Please reconsider the 

>context.

[Answer] 

We deleted the notation about ovarian cancer.

>9. Please consider code sharing as stated in https://journals.plos.org/plosone/s/materials-software-and-code-sharing.

[Answer] 

We attached the code of Gillespie algorithm simulation.

Minor points

>p.16 line 245 and p.19 line 305: The value of N_L seems to be 300 instead of 100, as the maximum 

>value of N in Fig 2 is 300.

[Answer] 

We did new simulation with N=100 and redrew Fig.2 as response to Major-comment 3. 

>p.20 line 314: The ministry name should be followed by the country name.

[Answer] 

The notation of the reference was specified, so we wrote the order as it is.

[Answer] 

We have corrected all the following concerns pointed out in the minor points. 

>p.3 lines 31-32 and other places: Please consider adding the word “previous” when referring to previous observations. 

>Example: … the estimated parameters did not always agree with “previously reported values”. (p.3 lines 31-32)

>p.47 line 47: “appearance” should be “mortality” according to ref. 2.

>p.6 lines 74-75: Consider the following modification: “… and ionizing radiation contributes to the carcinogenic 

>process by adding a few mutations”

>p.6 line 91: … due “to” replication errors …

>p.7 line 106, p.24 line 387, p.31 line 503: Replace the first comma (,) with a semicolon (;) (e.g., …; however, …).

>p.12 lines 184-185: “…, in the growth phase, …” This phrase should be deleted.

>p.12 line 186: The subscript g should be substituted with s (Gamma_g to Gamma_s).

>p.13 line 207 and other occasions: Because “oncogenes” do not include “tumor suppressor genes”, the wording here 

>should be “cancer driver genes” instead of “oncogenes”. The same applies to many occasions in the manuscript.

>p.14 line 217: “… of having more than g mutations …” should be “… of having g or more mutations …”

>p.14 line 221: “cancer risk” should be “cumulative cancer incidence”

>p.16 line 252: n should be d?

>p.17 line 273: the other -> another

>p.19 line 308 and other occasions: The wording of “statistical data” is odd and should be replaced with expressions 

>like “epidemiology data”, “real-world data” or “cancer registry data”.

>p.20 line 328: “value” -> “values”

>p.24 line 385: “larger” -> “smaller”?

>p.29 line 480: “with on time” -> “with time”?

>p.30 lines 501-502: The grammar of “… should be depends on …” needs reconsideration.

---

## [Decision Letter · Decision Letter 2]

20 Feb 2023

PONE-D-22-12989R2A mathematical model for cancer risk and accumulation of mutations caused by replication errors and external factorsPLOS ONE

Dear Dr. Uchonomiya:

Thank you for submitting your manuscript to PLOS ONE. After careful consideration, we feel that it has merit but does not fully meet PLOS ONE’s publication criteria as it currently stands. Therefore, we invite you to submit a revised version of the manuscript that addresses the points raised during the review process.

Minor changes mostly editorial in nature have been proposed for the work.  Please address these comments in a revision.

We look forward to receiving your revised manuscript.

Kind regards,

Gayle E. Woloschak, PhD

Section Editor

PLOS ONE

Journal Requirements:

Reviewers' comments:

Reviewer's Responses to Questions

**Comments to the Author**

1. If the authors have adequately addressed your comments raised in a previous round of review and you feel that this manuscript is now acceptable for publication, you may indicate that here to bypass the “Comments to the Author” section, enter your conflict of interest statement in the “Confidential to Editor” section, and submit your "Accept" recommendation.

Reviewer #2: All comments have been addressed

Reviewer #3: All comments have been addressed

2. Is the manuscript technically sound, and do the data support the conclusions?

Reviewer #2: Yes

Reviewer #3: Yes

3. Has the statistical analysis been performed appropriately and rigorously? 

Reviewer #2: Yes

Reviewer #3: Yes

4. Have the authors made all data underlying the findings in their manuscript fully available?

Reviewer #2: Yes

Reviewer #3: Yes

5. Is the manuscript presented in an intelligible fashion and written in standard English?

Reviewer #2: Yes

Reviewer #3: Yes

6. Review Comments to the Author

Reviewer #2: The authors have adequately addressed referee remarks. The study is ready for publication in my opinion.

Reviewer #3: The authors have adequately addressed my comments, with following minor points left. I apology that some of these points had been missed in the previous round of review.

1. Line 39: re-estimate -> re-estimated

2. Eq 1, 2, 6, 7, 12, 13, 15, 16 and 17: lambda_2 is used in some places instead of lambda_1.

3. Eq 22: i is used instead of j.

4. Line 269: “These means” -> “This means” or “These mean”

5. Line 303: ... the number of lesion“s” ...

7. PLOS authors have the option to publish the peer review history of their article (what does this mean?). If published, this will include your full peer review and any attached files.

Reviewer #2: No

Reviewer #3: No

---

## [Author Response · Author response to Decision Letter 2]

30 Apr 2023

>Reviewer #2: The authors have adequately addressed referee remarks. The study is ready for publication 

>in my opinion.

[Answer] We were relieved that our response seemed to address your concerns.

>Reviewer #3: The authors have adequately addressed my comments, with following minor 

>points left. I apology that some of these points had been missed in the previous round of 

>review.

>1. Line 39: re-estimate -> re-estimated

>2. Eq 1, 2, 6, 7, 12, 13, 15, 16 and 17: lambda_2 is used in some places instead of lambda_1.

>3. Eq 22: i is used instead of j.

>4. Line 269: “These means” -> “This means” or “These mean”

>5. Line 303: ... the number of lesion“s” ...

[Answer] Thank you very much for your careful checking. We have corrected all the points you pointed out.

---

## [Editor Report · Decision Letter 3]

18 May 2023

A mathematical model for cancer risk and accumulation of mutations caused by replication errors and external factors

PONE-D-22-12989R3

Dear Dr. Uchinomiya:

We’re pleased to inform you that your manuscript has been judged scientifically suitable for publication and will be formally accepted for publication once it meets all outstanding technical requirements.

Kind regards,

Gayle E. Woloschak, PhD

Section Editor

PLOS ONE
---

## [Editor Report · Acceptance letter]

22 May 2023

PONE-D-22-12989R3 

A mathematical model for cancer risk and accumulation of mutations caused by replication errors and external factors 

Dear Dr. Uchinomiya:

I'm pleased to inform you that your manuscript has been deemed suitable for publication in PLOS ONE. Congratulations! Your manuscript is now with our production department. 

Kind regards, 

on behalf of

Dr. Gayle E. Woloschak 

Section Editor

PLOS ONE